# FRESHLLMS:
# REFRESHING LARGE LANGUAGE MODELS WITH SEARCH ENGINE AUGMENTATION

## ABSTRACT

Most large language models (LLMs) are trained once and never updated; thus, they lack the ability to dynamically adapt to our ever-changing world. In this work, we perform a detailed study of the factuality of LLM-generated text in the context of answering questions that test current world knowledge. Specifically, we introduce FRESHQA, a novel dynamic QA benchmark encompassing a diverse range of question and answer types, including questions that require *fast-changing* world knowledge as well as questions with *false premises* that need to be debunked. We benchmark a diverse array of both closed and open-source LLMs under a two-mode evaluation procedure that allows us to measure both correctness and hallucination. Through human evaluations involving more than 50K judgments, we shed light on limitations of these models and demonstrate significant room for improvement: for instance, all models (regardless of model size) struggle on questions that involve fast-changing knowledge and false premises. Motivated by these results, we present FRESHPROMPT, a simple few-shot prompting method that substantially boosts the performance of an LLM on FRESHQA by incorporating relevant and up-to-date information retrieved from a search engine into the prompt. Our experiments show that FRESHPROMPT outperforms both competing search engine-augmented prompting methods such as SELF-ASK (Press et al., 2022) as well as commercial systems such as PERPLEXITY.AI.[1] Further analysis of FRESHPROMPT reveals that both the number of retrieved evidences and their order play a key role in influencing the correctness of LLM-generated answers. Additionally, instructing the LLM to generate concise and direct answers helps reduce hallucination compared to encouraging more verbose answers. To facilitate future research, we will release FRESHQA after blind review and commit to updating it at regular intervals.

## 1 INTRODUCTION

Recent large language models (LLMs) such as BARD and CHATGPT/GPT-4[2] are designed to be versatile open-domain *chatbots* that can engage in multi-turn conversations on diverse subjects. Despite their impressive capabilities, these LLMs often "hallucinate" plausible but factually incorrect information (Maynez et al., 2020; Liu et al., 2023b), which reduces the trustworthiness of their responses, especially in settings where accurate and up-to-date information is critical. This behavior can be partially attributed to the presence of outdated knowledge encoded in their parameters. While additional training using human feedback (Ouyang et al., 2022) or knowledge-enhanced tasks can mitigate this issue, it is not easily scalable for real-time knowledge updates (e.g., stock price of a company). In-context learning (Brown et al., 2020) is an appealing alternative in which real-time knowledge can be injected into an LLM's prompt for conditioning generation. While recent work has begun to explore augmenting LLMs with web search results (Lazaridou et al., 2022; Press et al., 2022), it is unclear how to take full advantage of search engine outputs to increase LLM factuality.

In this work, we collect a novel QA benchmark, dubbed FRESHQA, to evaluate the factuality of existing LLMs. FRESHQA consists of 600 natural questions that are broadly divided into the *four*

---

[1] https://www.perplexity.ai
[2] https://bard.google.com, https://chat.openai.com

| Type | Question | Answer (as of this writing) |
|---|---|---|
| never-changing | *Has Virginia Woolf's novel about the Ramsay family entered the public domain in the United States?* | ***Yes***, *Virginia Woolf's 1927 novel To the Lighthouse entered the public domain in 2023.* |
| never-changing | *What breed of dog was Queen Elizabeth II of England famous for keeping?* | ***Pembroke Welsh Corgi*** *dogs.* |
| slow-changing | *How many vehicle models does Tesla offer?* | *Tesla offers **five** vehicle models: Model S, Model X, Model 3, Model Y, and the Tesla Semi.* |
| slow-changing | *Which team holds the record for largest deficit overcome to win an NFL game?* | *The record for the largest NFL comeback is held by the **Minnesota Vikings**.* |
| fast-changing | *Which game won the Spiel des Jahres award most recently?* | ***Dorfromantik*** *won the 2023 Spiel des Jahres.* |
| fast-changing | *What is Brad Pitt's most recent movie as an actor* | *Brad Pitt recently starred in **Babylon**, directed by Damien Chazelle.* |
| false-premise | *What was the text of Donald Trump's first tweet in 2022, made after his unbanning from Twitter by Elon Musk?* | *He **did not tweet** in 2022.* |
| false-premise | *In which round did Novak Djokovic lose at the 2022 Australian Open?* | *He **was not allowed to play** at the tournament due to his vaccination status.* |

Figure 1: FRESHQA exemplars. Our questions are broadly divided into *four* main categories based on the nature of the answer: *never-changing*, in which the answer almost never changes; *slow-changing*, in which the answer typically changes over the course of several years; *fast-changing*, in which the answer typically changes within a year or less; and *false-premise*, which includes questions whose premises are factually incorrect and thus have to be rebutted.

main categories shown in Figure 1. FRESHQA's questions span a diverse set of topics with diverse difficulty levels (requiring single-hop and multi-hop reasoning), and require a model to "understand" the world's up-to-date knowledge to be able to answer correctly. Additionally, FRESHQA is dynamic in nature: some of the ground-truth answers may change over time, and a question classified under a specific category may undergo reclassification at some later point in time (e.g., the current *false-premise* question *"How long has Elon Musk been married to his current spouse?"* will fall into the *fast-changing* category if Elon Musk gets married again in the future).

We benchmark how well different LLMs perform on FRESHQA by prompting them with questions and optionally a few question-answer demonstrations and then sampling a response. Then, we conduct an extensive human evaluation of the factual accuracy of the models' responses, consisting of more than 50K judgements. We evaluate each response in a two-mode evaluation procedure: RELAXED, which measures only whether the main answer is correct; and STRICT, which measures whether all of the claims in the response are factual and up-to-date (i.e., no hallucination). Our study sheds light on the factuality of old and new LLMs and reveals different model behaviors across question types. Unsurprisingly, there are flat scaling curves on questions that involve fast-changing knowledge: simply increasing the model size does not lead to reliable performance gains. We also observe similar trends on false-premise questions, though several LLMs are able to debunk a false-premise question if explicitly asked *"Please check if the question contains a valid premise before answering"*. Overall, FRESHQA is challenging for current LLMs and leaves ample room for improvement.

Motivated by these findings, we further investigate how to effectively improve LLMs' factuality by grounding their responses to accurate and up-to-date information from search engines. Given the rapid development of ever larger LLMs and the ever-changing nature of knowledge, we explore in-context learning approaches that allow an LLM to attend over knowledge provided at inference time through its prompt. We develop FRESHPROMPT, a simple yet effective method that, for a given question, takes full advantage of a search engine by extracting all up-to-date and relevant information (including knowledge from relevant questions that search users also ask) and uses few-shot in-context learning to teach a model to reason over retrieved evidences and figure out the right answer. We show that FRESHPROMPT significantly boosts LLMs's factuality: for example, our best GPT-4 + FRESHPROMPT variant yields an improvement of 32.6% and 49.0% accuracy over the vanilla GPT-4 on FRESHQA under RELAXED and STRICT, respectively. Since our method requires no additional training, it is flexible and applicable to a variety of scenarios.

Taken together, our key contributions include:

- We introduce a novel dynamic QA benchmark, FRESHQA, which features a diverse set of question and answer types, including questions whose answers may change over time

and questions whose premises are factually incorrect. We will make our dataset freely available and commit to updating the ground-truth answers at a regular schedule to encourage exploration of methods to improve LLMs' factuality.

- We benchmark a wide range of both closed and open-source LLMs on our dataset. Through an extensive and rigorous human evaluation study, we shed light on limitations of current LLMs: they struggle on fast-changing, false-premise, and multi-hop questions, and our two-mode evaluation captures increased hallucinations produced by techniques such as chain-of-thought prompting (Wei et al., 2022).

- We present FRESHPROMPT, a simple in-context learning method that can substantially boost an LLM's factuality compared to competing search-augmented approaches by effectively incorporating factual and up-to-date information from a search engine into the model's prompt. Furthermore, we perform a series of sensitivity and ablation analyses to better understand what facets of FRESHPROMPT contribute to its success.

## 2    FRESHQA

In this section, we address the growing need to assess LLM factuality by curating a novel QA benchmark, FRESHQA, with 600 questions that cover a wide spectrum of question and answer types.

### 2.1    DATA COLLECTION

We collected FRESHQA by recruiting both NLP researchers (including the authors and their colleagues) and online freelancers[3] to write questions of varying difficulty levels and topics whose answers may change based on new developments in the world. The annotators were shown a few exemplars of the four broad types of questions defined in Figure 1. Within each of these four categories, we ask annotators to write questions at two different difficulty levels: *one-hop*, where the question explicitly mentions all of the relevant information needed to answer it, and thus no additional reasoning is required (e.g., *"Who is the CEO of Twitter"*); and *multi-hop*, where the question requires one or more additional steps of reasoning in order to gather all of the relevant information needed to answer it (e.g., *"What is the total height of the tallest building in the world?"*). Annotators were encouraged to write questions that involve *fresh* knowledge (knowledge that has changed recently or new events) and appear *natural* (i.e., plausible for a real person to type into a search engine). For false-premise questions, we requested a brief explanation elucidating why the question is flawed.[4]

**Quality control:**    Upon obtaining the initial dataset, we conducted multiple thorough data cleaning and quality assessments. This involved manual review of each example to ensure well-formed questions, removal of duplicates and invalid questions (e.g., too easy or controversial), and verification of answers and supporting evidence URLs. We also manually collected supplementary valid answers for each question (e.g., different names of the same person, different date formats, etc.). To facilitate future answer updates, we excluded questions whose answers are likely to change more frequently than once per week, and additionally incorporated the expected next review date for each question.

**Data size and split:**    The resulting dataset is divided into a *test* set consisting of 125 questions for each of the four broad question types (500 total examples) and a *development* set comprising 25 questions for each question type (100 total examples), sampled randomly within types. Additionally, 15 examples spanning different question types were extracted for *demonstration* purposes (i.e., for use in few-shot in-context learning), and the remaining data was discarded. The development set is reserved for future studies and not used in this paper.[5]

**FRESHQA requires regular updates:**    Our dataset has time sensitivity since the ground-truth answers may change with new developments in the world. As such, we commit to updating the dataset regularly and encourage researchers to evaluate on the latest version of the dataset, as close to the release date of the updated dataset as possible.

---

[3]We use UPWORK (https://www.upwork.com) with a compensation rate of $2 per example.

[4]Additionally, the annotators were asked to include the year the answer to the question last changed and an URL to a reputable website that supports the answer.

[5]Although our test set is currently balanced across question types, the distribution may change over time due to reclassification of questions from one category to another.

## 2.2 EVALUATION

All model responses were evaluated by the authors in a two-mode evaluation procedure: RELAXED, which focuses solely on evaluating the correctness of the primary answer; and STRICT, which additionally examines whether *all* of the facts in the answer are accurate (i.e., no hallucination). Overall, our setup provides both ends of the spectrum for evaluating factuality (the difference between a model's strict and relaxed performance provides a way to measure hallucination), offering a more comprehensive and nuanced understanding of their performance.

**Evaluation protocol:** In both evaluation modes, we credit a model's response only if it provides a confident and definitive answer, or the correct answer can be obviously inferred from the response. The primary or final answer when standing alone must be accurate. Any additional information that is provided must not contradict the primary answer or reshape one's perception of it. For false-premise questions, the model must point out the presence of a false premise to receive credit. For answers that involve names of entities (e.g., people), complete names or commonly recognized names are expected. Regarding numerical answers, approximate numbers are generally not accepted unless explicitly included in the ground-truth answers. Under RELAXED, we accept ill-formed responses (including those in a non-English language), as well as hallucinated or outdated information that does not significantly impact the primary answer. Under STRICT, however, a response that contains any hallucination, no matter how minor, will not receive credit. Furthermore, we accept a response in STRICT when the model indicates that the information might be outdated (e.g., *"As of my knowledge cutoff date in September 2021"*) *only* if it is evident that the knowledge has not changed.[6] Figure 4 in Appendix A shows specific examples of each evaluation criteria.

**Inter-rater agreement and automatic evaluation:** Two authors independently evaluated a subset of 100 answers in both modes and had an agreement of 99% for RELAXED and 96% for STRICT, showing that the protocol is reliable for comparing different LLMs. Additionally, to facilitate future evaluations, we develop FRESHEVAL, a simple automatic metric that uses few-shot in-context learning to teach an LLM to judge model responses, achieving an average agreement of 96.5% with human evaluations for RELAXED and 96% for STRICT. See Appendix B for details.

## 3 PRE-TRAINED LLMs STRUGGLE ON FRESHQA

We use FRESHQA to benchmark LLMs that do not have access to real-time data or the ability to browse the Internet for current information.[7] While all LLMs (regardless of size) predictably struggle on questions requiring up-to-date knowledge, they also underperform on false premise questions. In our experiments, we simply feed individual questions as prompts into each model and decode the model's predictions using a temperature of 0 without fine-tuning (see Appendix C for more details).

**Baselines:** We experiment with a series of models varying in size from 770M to 540B parameters, including basic pre-trained models such as **T5** (Raffel et al., 2020; Lester et al., 2021), **PALM** and **PALMCHILLA** (Chowdhery et al., 2022), optionally using **FEW-SHOT** prompting (Brown et al., 2020) and Chain-of-Thought (**COT**, Wei et al., 2022);[8] instruction-tuned models including **FLAN-T5** and **FLAN-PALM** (Chung et al., 2022; Longpre et al., 2023), and OPENAI's **GPT-3.5** (Ouyang et al., 2022), **CODEX** (Chen et al., 2021a), **CHATGPT**, and **GPT-4** (OpenAI, 2023).

### 3.1 RESULTS AND DISCUSSION

**FRESHQA presents a challenge for LLMs:** We visualize the accuracy of different LLMs on FRESHQA in both evaluation modes in Figure 2.[9] A first obvious takeaway is that all models struggle

---

[6]Note that even without access to real-time data, a model may still provide accurate answers to certain questions involving current information, potentially through random guesses or by leveraging past valid responses (e.g., for the question *"Which drama series won the most recent Primetime Emmy Award for Outstanding Drama Series?"*, while *"Succession"* won the award most recently (as of this writing), it was also the winner in 2020, so a model trained in 2021 could potentially provide the correct answer).

[7]With the exception of CHATGPT and GPT-4, which have access to the current date. Note that the latest versions of these models can now browse the Internet.

[8]As we are interested in exploring how these methods perform without being specifically designed for FRESHQA, we use the 5-shot demonstrations for TRIVIAQA (Joshi et al., 2017) used in Sun et al. (2023).

[9]Table 3 and Table 4 in Appendix D contain concrete numbers under STRICT and RELAXED, respectively.

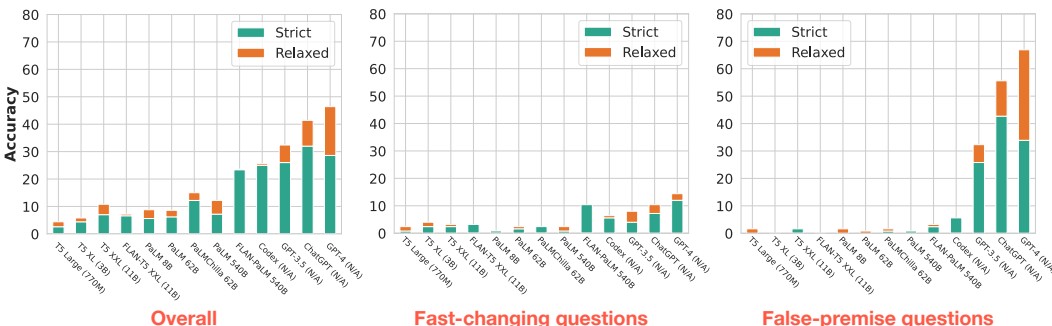

Figure 2: Accuracy of different LLMs on FRESHQA under RELAXED and STRICT (no hallucination) evaluations. Models benchmarked on the same date of April 26, 2023. *All* models (regardless of model size) struggle on questions that involve *fast-changing* knowledge and *false premises*.

on FRESHQA: overall accuracy ranges from 0.8% to 32.0% under STRICT, and 0.8% to 46.4% under RELAXED. Switching from RELAXED to STRICT results in a marked decrease in accuracy for CHATGPT and GPT-4. This is mainly due to the lack of access to up-to-date information, as they produce "outdated" answers (which often start with the prefix '*As of my knowledge cutoff date in September 2021*"), and in many cases, "refuse" to provide an answer (e.g., *"As an AI language model, I cannot provide real-time information."*). Similarly, the accuracy of PALM (across model sizes) drops significantly under STRICT. Much of this drop is due to artifacts such as conversation-like responses with unexpected special tokens (e.g., the end-of-turn [eot]), and hallucination. In contrast, FLAN-PALM and CODEX exhibit minimal hallucination due to their concise and direct answers.

**LLMs struggle with questions about current information:** The lack of up-to-date parametric knowledge results in dramatically degraded accuracies across models on questions involving fast-changing or recent knowledge. GPT-4 generally obtains the highest accuracy on these questions, with the exception of questions about recent knowledge (i.e., since 2022) under STRICT where it underperforms FLAN-PALM and CODEX, but it never exceeds 15% across both evaluation modes. Our evaluation confirms that CHATGPT and GPT-4 have been exposed to data containing information beyond their knowledge cutoff date (Appendix E). Additionally, GPT-4 is more reluctant to answer fast-changing questions (refusing to answer 60% of the time) compared to CHATGPT (16%).

**Questions with false premises pose a hurdle for LLMs:** All models struggle on questions with false premises, and using larger models does not increase accuracy for T5 and PALM ("flat scaling"), with performance within the range of 0.0% to 1.6%. GPT-3.5, CHATGPT, and GPT-4 demonstrate much superior accuracies to all other models, achieving accuracies between 25.8% to 42.7% under STRICT and 32.3% to 66.9% under RELAXED. CHATGPT performs the best under STRICT (42.7%) while GPT-4 is the most accurate model under RELAXED (66.9%), with an impressive accuracy of 83.9% on questions about knowledge before 2022. These results suggest that OPENAI's models are likely trained to cope with false-premise questions.

**CoT increases hallucination:** Overall, FEW-SHOT and CoT prompting are beneficial for large models and sometimes advantageous for moderately-sized models on questions with valid premises, especially on questions about never-changing or old knowledge. Under STRICT, FEW-SHOT and CoT yields +36.1% and +26.9% respective accuracy improvement over zero-shot prompting with PALM 540B on questions involving knowledge before 2022 (+21.9% and +29.7% under RELAXED). CoT largely demonstrates superior performance compared to FEW-SHOT under RELAXED, whereas FEW-SHOT obtains better results under STRICT, as CoT introduces more room for hallucination.

**Multi-hop reasoning is challenging for several models:** T5 LARGE and XL are incapable of dealing with multi-hop questions, while FLAN-PALM 540B, CODEX, and GPT-3.5 suffer the most when switching from one-hop to multi-hop questions. GPT-4 remains stable across these two types of questions (with a difference of less than 2% in accuracy across settings). See Appendix D for details.

## 4 PROMPTING SEARCH ENGINE-AUGMENTED LANGUAGE MODELS

The low accuracies reported in the previous section are largely unsurprising, as none of the models we evaluated had access to real-time information. In this section, we evaluate the impact of *search*

```
source: {source_webpage}          {demonstrations} # details omitted for brevity
date: {publication_date}
title: {title}                    query: {question}
snippet: {text_snippet}           {retrieved_evidences} # chronological order
highlight:                        question: {question}
{highlighted_words}               answer: {reasoning_and_answer}
```

Figure 3: FRESHPROMPT's format. We cast all retrieved evidences into a unified format with useful information, including source webpage, date, title, text snippet, and highlighted words (left). Few-shot demonstrations are provided at the beginning of the prompt. Each demonstration shows the model an example question and a list of retrieved evidences for the question, followed by some reasoning over the evidences to figure out the most relevant and up-to-date answer (right).

*engine augmentation* to LLMs on FRESHQA. We present FRESHPROMPT, a simple few-shot prompting method that substantially boosts FRESHQA performance of an LLM by incorporating relevant and up-to-date information retrieved from a search engine (GOOGLE SEARCH) into the prompt.

## 4.1 FRESHPROMPT

Our FRESHPROMPT method leverages a text prompt to (1) introduce contextually relevant and up-to-date information (including answers to relevant questions) from a search engine to a pre-trained LLM, and (2) teach the model to reason over retrieved evidences. More specifically, given a *question $q$*, we first use $q$ verbatim to query a search engine, in our case GOOGLE SEARCH.[10] We retrieve all of the search results, including the *answer box*, *organic results*, and other useful information, such as the *knowledge graph*, *questions and answers* from crowdsourced QA platforms, and *related questions* that search users also ask (see Figure 9 in Appendix F). For each of these results, we extract the associated *text snippet $x$* along with other information, such as *source $s$* (e.g., WIKIPEDIA), *date $d$*, *title $t$*, *highlighted words $h$*, and then create a list of $k$ retrieved evidences $E = \{(s, d, t, x, h)\}$. These evidences are then cast into a common format (Figure 3, left) and used to condition the model through in-context learning. To encourage the model to focus on more recent evidences, in line with recent findings (Liu et al., 2023a), we sort the evidences $E$ in the prompt from oldest to newest.

To help the model to "understand" the task and the desired output, we provide few-shot demonstrations of input-output exemplars at the beginning of the input prompt. Each demonstration shows the model an example question and a list of retrieved evidences for the question, followed by a chain-of-thought reasoning over the evidences to figure out the most relevant and up-to-date answer (Figure 3, right). Although we include a few exemplars of questions with false premises in the demonstrations, we also experiment with an explicit false premise check in the prompt: *"Please check if the question contains a valid premise before answering"*. Figure 10 in Appendix G shows a realistic prompt.

## 4.2 EXPERIMENT SETUP

We closely follow the setup in Section 3 except in cases where we lack control over the model's decoding via an API (e.g., PERPLEXITY.AI). Some of the models we evaluate can potentially change over time, which presents a challenge to the reproducibility of our evaluation results; thus, we evaluate all models on the same date of April 26, 2023. In addition to GPT-3.5 and GPT-4, we evaluate **GOOGLE SEARCH** by simply querying GOOGLE SEARCH and using the answer in the answer box (if any) or the text snippet of the top-1 search result; **PERPLEXITY.AI (PPLX.AI)**, an answer engine that combines an LLM and a search engine to generate useful responses to users' queries;[11] and **SELF-ASK** (Press et al., 2022), a method that uses few-shot in-context learning to teach an LLM to decompose each question into simpler sub-questions that are answered via GOOGLE SEARCH.[12]

**FRESHPROMPT setup:** We apply FRESHPROMPT to both GPT-3.5 and GPT-4 by sequentially incorporating the following retrieved evidences into the input prompt: $o$ organic search results, $r$

---

[10]We scrape the results from GOOGLE SEARCH using SERPAPI (https://serpapi.com).

[11]https://www.perplexity.ai. At the time of evaluation, PPLX.AI was a combination of GPT-3.5 and BING SEARCH, and was able to provide both concise and detailed answers. We evaluated its concise answers.

[12]We use the few-shot prompt provided by SELF-ASK's authors and apply it to both GPT-3.5 and GPT-4. For simplicity, we evaluate solely the final answer from SELF-ASK, disregarding intermediate answers.

Table 1: Accuracy of different search engine-augmented LLMs on FRESHQA under STRICT (no hallucination) evaluations. Models benchmarked on the same date of April 26, 2023. We report accuracy across different categories of questions, including *fast-changing* (*fast*), *slow-changing* (*slow*), *never-changing* (*never*), false-premise, questions that involve knowledge before 2022 ($< 2022$) and since 2022 ($\geq 2022$), one-hop (*1-hop*) and multi-hop (*m-hop*) questions. $^+$ indicates a model with access to the current date. UTD stands for "up-to-date".

| Model (size) | knowl. cutoff | all | valid premise | | | | | | | | false premise | |
|---|---|---|---|---|---|---|---|---|---|---|---|---|
| | | | all | fast | slow | never | $< 2022$ | $\geq 2022$ | 1-hop | $m$-hop | all | $< 2022$ |
| *comparison against baselines* | | | | | | | | | | | | |
| GOOGLE SEARCH (N/A) | UTD | 39.6 | 48.9 | 32.0 | 46.4 | 68.3 | 67.4 | 37.9 | 55.6 | 32.4 | 11.3 | 9.7 |
| | | | | | | | | | | | | |
| GPT-3.5 (N/A) | 2021 | 26.0 | 26.1 | 4.0 | 15.2 | 58.7 | 61.0 | 5.1 | 28.0 | 21.3 | 25.8 | 34.4 |
| GPT-3.5 + SELF-ASK (N/A) | UTD | 41.6 | 51.1 | 36.8 | 43.2 | 73.0 | 73.8 | 37.4 | 52.2 | 48.1 | 12.9 | 17.2 |
| GPT-3.5 + FRESHPROMPT | UTD | 56.0 | 62.5 | 46.4 | 60.8 | 80.2 | 71.6 | 57.0 | 68.7 | 47.2 | 36.3 | 43.0 |
| PPLX.AI (N/A) | UTD | 52.2 | 57.2 | 38.4 | 53.6 | 79.4 | 73.0 | 47.7 | 63.8 | 40.7 | 37.1 | 38.7 |
| | | | | | | | | | | | | |
| GPT-4 (N/A) | 2021$^+$ | 28.6 | 26.9 | 12.0 | 4.0 | 64.3 | 58.2 | 8.1 | 27.2 | 25.9 | 33.9 | 41.9 |
| GPT-4 + SELF-ASK (N/A) | UTD | 47.8 | 47.1 | 39.2 | 46.4 | 55.6 | 51.8 | 44.3 | 43.7 | 55.6 | 50.0 | 61.3 |
| GPT-4 + FRESHPROMPT | UTD | **75.6** | **77.1** | **59.2** | **77.6** | **94.4** | **88.7** | **70.2** | **81.3** | **66.7** | **71.0** | **77.4** |
| *sensitivity and ablation studies* | | | | | | | | | | | | |
| GPT-3.5 (N/A) | 2021 | 26.0 | 26.1 | 4.0 | 15.2 | 58.7 | 61.0 | 5.1 | 28.0 | 21.3 | 25.8 | 34.4 |
| GPT-3.5 + FRESHPROMPT | UTD | 56.0 | 62.5 | 46.4 | 60.8 | 80.2 | 71.6 | 57.0 | 68.7 | 47.2 | 36.3 | 43.0 |
| w/ PREMISE CHECK | UTD | 35.2 | 27.1 | 14.4 | 28.0 | 38.9 | 36.2 | 21.7 | 31.0 | 17.6 | 59.7 | 67.7 |
| | | | | | | | | | | | | |
| GPT-4 (N/A) | 2021$^+$ | 28.6 | 26.9 | 12.0 | 4.0 | 64.3 | 58.2 | 8.1 | 27.2 | 25.9 | 33.9 | 41.9 |
| | | | | | | | | | | | | |
| GPT-4 w/ SNIPPETS ONLY & SEARCH ORDER | UTD | 74.0 | 75.5 | 56.8 | 75.2 | 94.4 | 87.9 | 68.1 | 79.9 | 64.8 | 69.4 | 77.4 |
| GPT-4 w/ SNIPPETS ONLY & TIME ORDER | UTD | 74.8 | 75.5 | 58.4 | 74.4 | 93.7 | 87.9 | 68.1 | 79.9 | 64.8 | 72.6 | **82.8** |
| GPT-4 w/ SNIPPETS ONLY & RANDOM ORDER | UTD | 72.4 | 73.7 | 56.8 | 69.6 | 94.4 | 87.9 | 65.1 | 78.4 | 62.0 | 68.5 | 76.3 |
| | | | | | | | | | | | | |
| GPT-4 + FRESHPROMPT | UTD | 75.6 | 77.1 | 59.2 | 77.6 | 94.4 | **88.7** | 70.2 | 81.3 | 66.7 | 71.0 | 77.4 |
| w/ PREMISE CHECK | UTD | 75.0 | 74.2 | 56.8 | 76.0 | 89.7 | 85.1 | 67.7 | 79.5 | 61.1 | **77.4** | 79.6 |
| w/o ANSWER BOX | UTD | 74.2 | 74.7 | 57.6 | 74.4 | 92.1 | **88.7** | 66.4 | 79.1 | 63.9 | 72.6 | 78.5 |
| w/o ANSWER BOX & RELEVANT INFO | UTD | 72.4 | 72.9 | 54.4 | 71.2 | 92.9 | 87.2 | 64.3 | 78.0 | 60.2 | 71.0 | 78.5 |
| w/ 1 EVIDENCE | UTD | 61.4 | 60.9 | 40.0 | 55.2 | 87.3 | 79.4 | 49.8 | 66.8 | 46.3 | 62.9 | 75.3 |
| w/ 5 EVIDENCES | UTD | 70.6 | 72.1 | 56.0 | 69.6 | 90.5 | 81.6 | 66.4 | 78.0 | 57.4 | 66.1 | 73.1 |
| w/ 15 EVIDENCES | UTD | **77.6** | **78.5** | **60.8** | **78.4** | **96.0** | **88.7** | **72.3** | **81.7** | **70.4** | 75.0 | 80.6 |
| w/ 15 DEMONSTRATIONS | UTD | 74.6 | 75.5 | 56.8 | 76.0 | 93.7 | 87.9 | 68.1 | 79.9 | 64.8 | 71.8 | 76.3 |
| w/ LONG DEMONSTRATION ANSWERS | UTD | 73.0 | 72.6 | 55.2 | 71.2 | 91.3 | 83.7 | 66.0 | 77.6 | 60.2 | 74.2 | 81.7 |

related questions that search users also ask, $a$ questions and answers from crowdsourced QA platforms, and the snippets from the knowledge graph and answer box (if available). These evidences are arranged in sequence up to the end of the prompt. Given the models' context limit, we only keep the top $n$ evidences (closer to the end of the prompt) after sorting them based on the corresponding date. Unless otherwise specified, we use $(o, r, a, n, m) = (10, 2, 2, 5)$ for GPT-3.5, and $(o, r, a, n, m) = (10, 3, 3, 10)$ for GPT-4. Additionally, we include $m = 5$ question-answer demonstrations at the beginning of the prompt.

## 4.3 RESULTS AND DISCUSSION

**FRESHPROMPT significantly improves FRESHQA accuracy:** Table 1 presents concrete numbers under STRICT (see Appendix H for results under RELAXED). FRESHPROMPT offers large improvements over the vanilla GPT-3.5 and GPT-4 across the board. GPT-4 + FRESHPROMPT achieves absolute accuracy improvements of 47% and 31.4% over GPT-4 under STRICT and RELAXED, respectively. The reduction in the absolute accuracy gap between STRICT and RELAXED (from 17.8% to 2.2%) also suggests that FRESHPROMPT dramatically diminishes the presence of outdated and hallucinated answers. Unsurprisingly, the most significant improvements for both GPT-3.5 and GPT-4 are on the categories of fast-changing and slow-changing questions, which both concern recent knowledge. That said, questions about old knowledge also benefit from FRESHPROMPT. For example, GPT-4 + FRESHPROMPT yields a +30.5% higher accuracy than GPT-4 on questions with valid premises that involve knowledge before 2022 (+9.9% under RELAXED). Additionally, FRESHPROMPT produces notable gains on false-premise questions (+37.1% and +8.1% respective accuracy improvements under STRICT and RELAXED for GPT-4).

**FRESHPROMPT outperforms other search-augmented methods by a large margin:** GPT-4 + FRESHPROMPT demonstrates superior accuracy across question types, surpassing all other methods by a substantial margin. Its best variant (with 15 retrieved evidences per question) achieves impressive overall accuracies of 77.6% and 79.0% under STRICT and RELAXED, respectively. GPT-3.5 + FRESHPROMPT surpasses PPLX.AI and SELF-ASK (all performed on top of GPT-3.5) in overall accuracy by +3.8% and +14.4% under STRICT. Under RELAXED, however, PPLX.AI achieves a +4.2% higher

overall accuracy than GPT-3.5 + FRESHPROMPT, which is a large part due to its superior accuracy on false-premise questions (58.1% vs. 41.1%). The large accuracy gap of 14.0% between STRICT and RELAXED for PPLX.AI suggests that its outputs contain a large amount of hallucination. Overall, all search-engine augmented approaches (SELF-ASK, PPLX.AI, and FRESHPROMPT) provide significant gains across question types over vanilla GPT-3.5 and GPT-4. GOOGLE SEARCH generally provides better results than both GPT-3.5 and GPT-4, except on questions with false premises, but lags far behind PPLX.AI and GPT-3.5/GPT-4 + FRESHPROMPT across the board.

**The premise check boosts accuracy on false-premise questions but can hurt accuracy on those with valid premises:**  As discussed in Section 3.1, OPENAI's LLMs such as GPT-3.5 and GPT-4 are likely tuned to handle false-premise questions, and this is also true for PPLX.AI. Additionally, we empirically find that several LLMs possess the ability to debunk a false-premise question if explicitly asked, e.g.. *"Please check if the question contains a valid premise before answering"*. Adding this premise check to GPT-3.5 and GPT-4 yields +23.4% and +6.4% respective accuracy improvement on false-premise questions under STRICT (+22.6% and +11.3% under RELAXED). However, this is harmful for GPT-3.5 with regard to other question types, decreasing overall accuracy by 20.8% and 21% under STRICT and RELAXED, respectively. This is not a problem for GPT-4, with a slight decrease of 0.6% under STRICT and a slight increase of and 1.2% under RELAXED.

**Having more relevant and up-to-date evidences at the end of the input context is helpful:**  We also analyze how the order of the evidences in the prompt impacts GPT-4's accuracy. Our results show that using the order returned by GOOGLE SEARCH (SEARCH ORDER, top search results at the end of the input context) or sorting the evidences by their associated date information (TIME ORDER, more recent results at the end) generally results in better accuracy compared to using a random order (RANDOM ORDER), with up to a +2.2% higher overall accuracy in STRICT and RELAXED. Using only the text snippet for each evidence without additional information (such as source, date, etc.) as in GPT-4 + FRESHPROMPT slightly reduces accuracy, with less than 1% in both settings.

**Additional retrieved information beyond the organic search results provides further gains:**  Incorporating additional retrieved evidences other than the *organic search results*, such as the *answer box* or *related questions* that search users also ask, is helpful. Removing the *answer box* decreases GPT-4 + FRESHPROMPT's overall accuracy under STRICT by 1.4% (1.6% under RELAXED). Removing both the *answer box* and other relevant information (including *related questions*) reduces GPT-4 + FRESHPROMPT's overall accuracy by 3.2% (3.0% under RELAXED).

**Increasing the number of retrieved evidences further improves FRESHPROMPT:**  We explore the effect of the number of retrieved evidences for each question as well as the number of demonstrations by varying these numbers in our experiments with GPT-4. Note that our default setting for GPT-4 + FRESHPROMPT uses 10 retrieved evidences for each question and 5 demonstrations. Our results suggest that the number of retrieved evidences for each question is the most important ingredient for achieving highest accuracy. Under STRICT, increasing this number from 1 to 5, 10, and 15 leads to corresponding overall accuracy improvements of +9.2%, +14.2%, and +16.2%, respectively. This suggests that GPT-4 is able to efficiently handle an increasing number of retrieved evidences (including conflicting answers) and ground its responses into the most factual and up-to-date information. On the other hand, increasing the number of *demonstrations* from 5 to 15 slightly hurts accuracy in both evaluation settings (1% decrease in overall accuracy under STRICT).

**Verbose demonstrations improve on complex questions but also increase hallucination:**  To evaluate the effect of the writing style of the answer (including the reasoning) in each demonstration, we manually rewrite these answers into a more verbose version (LONG DEMONSTRATION ANSWERS). Our manual inspection reveals that using more verbose demonstration answers may be helpful when dealing with complex questions but can be more harmful as it provides room for hallucination (a decrease of 2.6% in overall accuracy under STRICT).

## 5 RELATED WORK

**Knowledge augmented LLMs:**  Many prior works study semi-parametric knowledge augmentation in LLMs via additional fine-tuning (Guu et al., 2020; Lewis et al., 2020; Borgeaud et al., 2022; Izacard et al., 2022), while others advocate for knowledge generation instead of retrieval (Yu et al., 2023a; Sun et al., 2023). FRESHPROMPT aligns with a recent emerging trend in QA applications that augments LLMs' prompts with knowledge retrieved from search engines for real-time alignment to current and factual information (Nakano et al., 2021; Lazaridou et al., 2022; Menick et al., 2022; Yao et al., 2022;

Press et al., 2022; Khattab et al., 2022; Schick et al., 2023; Luo et al., 2023). Similar to our method, Lazaridou et al. (2022) proposed a few-shot in-context learning approach that inserts documents from GOOGLE SEARCH into the prompt. We do not compare to this method due to its expensive inference cost, as it chunks retrieved documents into evidence paragraphs and performs $k = 50$ inference calls to the LLM to generate $k$ answers followed by LLM reranking. In contrast, FRESHPROMPT only performs a single inference call to the LLM. SELF-ASK (Press et al., 2022) also uses few-shot in-context learning to teach an LLM to ask itself follow-up questions before answering the initial question, although it focuses more on decomposition.

**Time-sensitive QA:** FRESHQA aligns with a growing body of work on benchmarking LLMs' temporal reasoning capabilities (Chen et al., 2021b; Zhang & Choi, 2021; Liska et al., 2022; Kasai et al., 2022). Chen et al. (2021b) created TIMEQA by extracting evolving facts from WIKIDATA along with aligned WIKIPEDIA passages to synthesize 20K timestamped question-answer pairs. Zhang & Choi (2021) constructed SITUATEDQA by annotating 9K realistic questions from existing open-domain QA datasets with temporal context (i.e., timestamps). STREAMINGQA (Liska et al., 2022) consists of both LLM-generated and human-written questions (146K total questions) answerable from a corpus of timestamped news articles. Also related is the dynamic REALTIMEQA benchmark (Kasai et al., 2022), which evaluates models weekly on a set of around 30 multiple-choice questions about new events extracted from news websites. In contrast, FRESHQA contains a fixed set of human written open-ended questions whose answers by nature can change based on new developments in the world and thus offers a complementary generative evaluation of time-sensitive QA.

**QA over questionable or counterfactual premises:** Recent work has also introduced QA benchmarks with questionable premises (Yu et al., 2023c; Kim et al., 2023) or counterfactual premises (Yu et al., 2023b). CREPE (Yu et al., 2023c) consists of 8400 Reddit questions (of which 25% questions contain false premises annotated by human workers) split into train/dev/test sets. Kim et al. (2023) constructed $(QA)^2$, an evaluation set of 602 questions based on frequent search engine queries, which are annotated by expert annotators and crowdworkers, and evenly divided between those with and without questionable premises. Consistent with these efforts, we find that current LLMs struggle with handling false premise questions; additionally, several LLMs are able to debunk a false-premise question if explicitly asked to check for the premise's validity. Similar to above, these benchmarks are complementary and combining them is a promising direction for future work.

## 6 LIMITATIONS AND FUTURE WORK

One obvious challenge with FRESHQA is the need for regular answer updating by the maintainers; in the interim period between updates, the answers to some questions might become stale. This could be addressed by support from the open-source community (e.g., updates via GITHUB pull requests). On the method side, FRESHPROMPT interfaces with GOOGLE SEARCH, and it is unclear how it performs with other search engines for which some types of context (e.g., answer boxes) are not available. Additionally, we only perform one search query per question, and thus our method could be further improved via question decomposition and multiple search queries (Khattab et al., 2022). Since FRESHQA consists of relatively simple English language questions, it is also unclear how well FRESHPROMPT performs in the context of multilingual/cross-lingual QA and long-form QA (Fan et al., 2019). Finally, FRESHPROMPT relies on in-context learning and thus may underperform approaches that fine-tune the base LLM on new knowledge.

## 7 CONCLUSION

Our work offers a fine-grained and exhaustive evaluation of the capabilities of modern LLMs to adapt to ever-changing world knowledge with and without search engine augmentation. In the process, we develop a new dataset—FRESHQA—of 600 questions that test a broad range of reasoning abilities, from the incorporation of fast-changing knowledge to identification of questions with false premises. Our two-mode evaluation also provides a way to measure both correctness and hallucination. Additionally, we propose a simple few-shot in-context learning algorithm called FRESHPROMPT that incorporates relevant evidences retrieved from GOOGLE SEARCH into the prompt of an LLM. FRESHPROMPT significantly improves performance over competing search engine-augmented approaches on FRESHQA, and an ablation reveals that factors such as the number of incorporated evidences and their order impact the correctness of LLM-generated answers. We release FRESHQA and commit to updating its answers regularly to facilitate future research.

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

APPENDIX

## A  EVALUATION PROTOCOL

Figure 4 shows specific examples of each evaluation criteria.

## B  INTER-RATER AGREEMENT AND AUTOMATIC EVALUATION

Two authors independently evaluated a randomly sampled subset of 100 answers across models (including 50 questions with valid premises and 50 questions with false premises) in both modes RELAXED and STRICT.

To facilitate future evaluations, we also develop FRESHEVAL, a simple automatic metric that uses few-shot in-context learning to teach an LLM to judge model responses. In each evaluation, the model is conditioned on a given question, a list of valid answers for the question, and a model response, and is then expected to generate a comment on the correctness of the response, followed by a final judgement. At the beginning of each input prompt, we also provide an instruction of the evaluation task, and sample comments and evaluations of the examples in Figure 4 as demonstrations.[13] See Figure 5 and Figure 6 for FRESHEVAL's prompts for RELAXED and STRICT evaluations, and Figure 7 for FRESHEVAL's sample output for STRICT evaluation.

Table 2 reports the inter-rater agreement between the two human raters, and between FRESHEVAL and each human rater, in terms of exact accuracy. The two human raters had an agreement of 99% for RELAXED and 96% for STRICT, while FRESHEVAL achieved an average agreement of 96.5% with human evaluations for RELAXED and 96% for STRICT. Overall, the high accuracies demonstrate that our evaluation protocol is reproducible and reliable, and FRESHEVAL can be used in place of human evaluation on FRESHQA.

## C  ADDITIONAL EXPERIMENT SETUP DETAILS FOR SECTION 3

To increase reproducibility, we select the most likely token at every decoding timestep (i.e., with a temperature of 0) and generate a maximum number of 256 tokens for all models. Note that the API for some models is non-deterministic by default, even with a temperature of 0. For non-chat models that were not pre-trained with a QA task, we feed them a text prompt of the format: "Q: <question>\nA: " ("\ n" is the new line character).

For OPENAI models, we use the `2023-03-15-preview` API in AZURE OPENAI SERVICE. We use the model names `text-davinci-003`, `code-davinci-002`, `gpt-3.5-turbo`, and `gpt-4` for GPT-3.5, CODEX, CHATGPT, and GPT-4, respectively.

## D  ADDITIONAL EXPERIMENT RESULTS FOR SECTION 3

Table 3 and Table 4 show the accuracy of different LLMs on FRESHQA under STRICT (no hallucination) and RELAXED evaluations, respectively.

## E  CHATGPT/GPT-4'S AWARENESS OF RECENT KNOWLEDGE

Although CHATGPT and GPT-4 were originally trained in 2021, our manual evaluation suggests that they have been exposed to data containing information beyond their knowledge cutoff date in September, 2021. Figure 8 indicates that CHATGPT is aware of the recent Russian invasion of Ukraine on February 24, 2022.

---

[13]In our experiments, we found that using separate prompts for RELAXED and STRICT evaluations resulted in better performance compared to using a single, combined prompt for both evaluation modes. We also found that additionally incorporating retrieved evidences for the question into the prompt did not improve inter-rater agreement between FRESHEVAL and human raters.

## F    GOOGLE SEARCH RESULTS

Figure 9 shows different types of search results from GOOGLE SEARCH for given a query.

## G    A REALISTIC PROMPT FOR FRESHPROMPT

Figure 10 displays a realistic prompt for FRESHPROMPT.

## H    ADDITIONAL EXPERIMENT RESULTS FOR SECTION 4

Table 5 presents the accuracy of different search engine-augmented LLMs on FRESHQA under RELAXED.

| ID | Question | Type | Primary answer (as of this writing) | Model response | Relaxed | Strict | Explanation |
|----|----------|------|-------------------------------------|----------------|---------|--------|-------------|
| 1 | How old is the world's oldest verified living person? | fast-changing | 116 years old | *The oldest verified living person is Maria Branyas Morera, who was born on March 4, 1907, making her 116 years old.* | ✓ | ✓ | the primary answer is correct, and all the provided information is accurate and up-to-date |
| 2 | When did the UK adopt the Euro? | false-premise | The UK has never adopted the Euro. | *The UK has never adopted the Euro as its official currency. The country has retained the British pound sterling (GBP) as its currency.* | ✓ | ✓ | the response debunks the question's false premise, and all the provided information is accurate and up-to-date |
| 3 | How much longer will Brittney Griner spend in the Russian prison? | false-premise | She was released in 2022. | I'm sorry, but *I have no information* to suggest that Brittney Griner is currently in a Russian prison. | ✗ | ✗ | a confident and definitive answer is required |
| 4 | Which language has the third largest population by number of native speakers? | slow-changing | English | *1. Mandarin 2. Spanish* **3. English** | ✓ | ✗ | though the correct answer can be obviously inferred from the response, a response without an explicit statement is less desirable |
| 5 | Has Virginia Woolf's novel about the Ramsay family entered the public domain in the United States? | never-changing | Yes | **1927's To the Lighthouse is in the public domain in the United States.** *The final answer is To the Lighthouse.* | ✗ | ✗ | the final answer when standing alone must be correct |
| 6 | Is Tesla's stock price above $200 a share? | fast-changing | Yes | *Tesla's stock price is above $200 a share. The stock is trading at a price of $120.* | ✗ | ✗ | conflicting information |
| 7 | When did Lionel Messi win his second FIFA World Cup? | false-premise | Messi has not won a second FIFA World Cup. | **This question does not contain a valid premise.** *Lionel Messi has never won a FIFA World Cup.* | ✗ | ✗ | despite debunking the question's false premise, the response inaccurately shapes one's knowledge perception |
| 8 | How many times has Jeremy Corbyn served as Prime Minister of the UK? | false-premise | Corbyn has never served as Prime Minister of the UK. | *Zero.* | ✗ | ✗ | a rebuttal is always required for false-premise questions |
| 9 | Who founded Amazon? | never-changing | Jeff Bezos | 100% of people said *Jeff*. | ✗ | ✗ | a complete name or commonly recognized name is required |
| 10 | How many books has Colleen Hoover published? | fast-changing | 25 | Colleen Hoover has published *over 20* books and novellas. | ✗ | ✗ | approximate numbers are not accepted unless included in the ground-truth answers |
| 11 | What is the largest model size of GPT-3? | never-changing | 175B parameters | **175 billion parameters.** *It is a 175 billion parameter, 1.37TB, 137B token, 137B wordpiece, 137B BPE, 137B byte pair encoding, 137B BPE token, 137B BPE wordpiece, 137B BPE token, 137B BPE wordpiece,...* | ✓ | ✗ | the primary answer is correct, but the response is ill-formed |
| 12 | How many member nations exist in the United Nations? | slow-changing | 193 | *As of my knowledge cutoff date in September 2021,* there were **193** member nations in the United Nations. | ✓ | ✗ | the primary answer is correct, but it is unclear whether the information is still up-to-date |
| 13 | Who won the biggest single-tournament payday in tennis history? | slow-changing | Novak Djokovic | The biggest single-tournament payday in tennis history was won by **Novak Djokovic** at the *2021* Australian Open. | ✓ | ✗ | the primary answer is correct, but the response contains hallucinated information |
| 14 | Who was the first football player to score one hundred international goals? | never-changing | Ali Daei | *As of my knowledge cutoff date in September 2021,* **Ali Daei**, a former Iranian footballer, *was the first* player to score 100 international goals. | ✓ | ✓ | all the information is correct and it is evident that the knowledge has not changed |
| 15 | Who is the latest winner of the Formula 1 world championship? | fast-changing | Max Verstappen | **Max Verstappen** won the 2022 Formula 1 world championship. He is a Dutch racing driver who competes in Formula One for Red Bull Racing. He is the son of former Formula One driver Jos Verstappen. Verstappen started his racing career in karting at the age of seven. He won the 2013 European Karting Championship and the 2013 World Karting Championship. In 2014, he moved to single-seater racing, competing in the European Formula Three Championship. *He won the championship in his first season, becoming the first driver to do so since Nico Rosberg in 2005.* In 2015, Verstappen moved to Formula One, driving for Toro Rosso. He became the youngest driver to compete in Formula One at the age of 17. He scored his first points in Formula One at the 2015 Hungarian GrandPrix. In 2016, Verstappen moved to Red Bull Racing. He won his first race at the 2016 Spanish Grand Prix. He became the youngest driver to win a race in Formula One at theage of 18. Verstappen finished the 2016 season in third place in the drivers' championship. In 2017, Verstappen won four races and finished the season in second place in the drivers' championship. In 2018, Verstappen won seven races and finished the season in second place in the drivers' championship. In 2019, Verstappen won nine races and finished the season in first place in the drivers' championship. He is the first Dutch driver to win the Formula One world championship. | ✓ | ✗ | the primary answer is correct, but the response contains several hallucinations (e.g., Max Verstappen did not win the Formula Three European Championship in 2014) |

Figure 4: FRESHQA sample evaluation. To get credit in both evaluation modes RELAXED and STRICT, all the information in the answer must be accurate and up-to-date (examples 1 and 2). In both modes, we credit a model's response only if it provides a confident and definitive answer (example 3), or the correct answer can be obviously inferred from the response (provided all other requirements are satisfied, see example 4). The primary or final answer when standing alone must be accurate (example 5). Any additional information that is provided must not contradict the primary answer (example 6) or reshape one's perception of it (example 7). For false-premise questions, the model must point out the presence of a false premise to receive credit (example 8). For answers that involve names of entities (e.g., people), complete names or commonly recognized names are expected (example 9). Regarding numerical answers, approximate numbers are generally not accepted unless explicitly included in the ground-truth answers (example 10). Under RELAXED, we accept ill-formed responses (including those in a non-English language), as well as hallucinated or outdated information that does not significantly impact the primary answer; under STRICT, however, a response that contains any hallucination, no matter how minor, will not receive credit (examples 11, 12, and 13). Furthermore, we accept a response in STRICT when the model indicates that the information might be outdated (e.g., *"As of my knowledge cutoff date in September 2021"*) *only* if it is evident that the knowledge has not changed (example 14).

Table 2: Inter-rater agreement between two authors (RATER 1 and RATER 2), and between FRESHEVAL and each human rater, in terms of exact accuracy across 100 RELAXED judgements, 100 STRICT judgements, and all ALL 200 judgements. In each of these three categories, in addition to the overall accuracy (**overall**), we report accuracy across questions with valid premises (**vp**) and questions with false premises (**fp**). The high accuracies demonstrate that our evaluation protocol is reproducible and reliable, and FRESHEVAL can be used in place of human evaluation on FRESHQA.

|  | RELAXED | | | STRICT | | | ALL | | |
| --- | --- | --- | --- | --- | --- | --- | --- | --- | --- |
|  | **overall** | **vp** | **fp** | **overall** | **vp** | **fp** | **overall** | **vp** | **fp** |
| RATER 1 vs. RATER 2 | **99.0** | **98.0** | **100** | 96.0 | **100.0** | 92.0 | **97.5** | **99.0** | **96.0** |
| FRESHEVAL vs. RATER 1 | 97.0 | **98.0** | 96.0 | **97.0** | **100.0** | **94.0** | 97.0 | **99.0** | 95.0 |
| FRESHEVAL vs. RATER 2 | 96.0 | 96.0 | 96.0 | 95.0 | **100.0** | 90.0 | 95.5 | 98.0 | 93.0 |

Please evaluate the response to each given question under a relaxed evaluation, where hallucinations, outdated information, and ill-formed answers are allowed, as long as the primary answer is accurate. Please credit the response only if it provides a confident and definitive answer, or the correct answer can be obviously inferred from the response. The primary or final answer when standing alone must be accurate. Any additional information that is provided must not contradict the primary answer or reshape one's perception of it. For false-premise questions, the response must point out the presence of a false premise to receive credit. For answers that involve names of entities (e.g., people), complete names or commonly recognized names are expected. Regarding numerical answers, approximate numbers are generally not accepted unless explicitly included in the ground-truth answers. We accept ill-formed responses (including those in a non-English language), as well as hallucinated or outdated information that does not significantly impact the primary answer.

```
# some demonstrations are omitted for brevity
```

**question:** Is Tesla's stock price above $250 a share?
**correct answer(s):** Yes
**response:** Yes, it is. The stock price is currently at $207.
**comment:** This is a valid question. While the primary answer in the response (Yes) is accurate, the additional information contradicts the primary answer ($207 is not above $250). Thus, the response is not credited.
**evaluation:** incorrect

**question:** Who is the latest winner of the Formula 1 world championship?
**correct answer(s):** Max Verstappen | Max Emilian Verstappen
**response:** Max Verstappen won the 2022 Formula 1 world championship. He is a Dutch racing driver who competes in Formula One for Red Bull Racing. He is the son of former Formula One driver Jos Verstappen. Verstappen started his racing career in karting at the age of seven. He won the 2013 European Karting Championship and the 2013 World Karting Championship. In 2014, he moved to single-seater racing, competing in the European Formula Three Championship. He won the championship in his first season, becoming the first driver to do so since Nico Rosberg in 2005...      `# the rest is omitted for brevity`
**comment:** This is a valid question. Even though the response contains several instances of hallucinated information (e.g., Max Verstappen did not win the Formula Three European Championship in 2014), the primary answer in the response (Max Verstappen) is still accurate. Thus, the response is credited.
**evaluation:** correct

**question:** In what year did bodybuilding become an Olympic sport?
**correct answer(s):** Bodybuilding is not an Olympic sport.
**response:** Bodybuilding is not currently an Olympic sport. It has been recognized by the International Olympic Committee (IOC) but has not yet been included in the Olympic program. However, bodybuilding has been included in other multi-sport events such as the Pan American Games and the Asian Games.
**comment:**

Figure 5: FRESHEVAL's prompt for RELAXED evaluation.

Please evaluate the response to each given question under a strict evaluation, where no hallucinations, outdated information, or ill-formed answers are allowed. Please credit the response only if it provides a confident and definitive answer, or the correct answer can be obviously inferred from the response. The primary or final answer when standing alone must be accurate. Any additional information that is provided must not contradict the primary answer or reshape one's perception of it. For false-premise questions, the response must point out the presence of a false premise to receive credit. For answers that involve names of entities (e.g., people), complete names or commonly recognized names are expected. Regarding numerical answers, approximate numbers are generally not accepted unless explicitly included in the ground-truth answers. A response that contains any hallucination, no matter how minor, will not receive credit. Furthermore, when the response indicates that the information might be outdated, we accept it only if it is evident that the knowledge has not changed (e.g., through common sense or well-known facts).

`# some demonstrations are omitted for brevity`

**question:** How old is the world's oldest verified living person?
**correct answer(s):** 116 years old | 116
**response:** As of today `{current_date}`, the most up-to-date and relevant information regarding this query is as follows. The oldest verified living person is Maria Branyas Morera, who was born on March 4, 1907, making her 116 years old.
**comment:** This is a valid question. The primary answer in the response (116 years old) is accurate, and all the provided information is accurate and up-to-date. Thus, the response is credited.
**evaluation:** correct

**question:** Who is the latest winner of the Formula 1 world championship?
**correct answer(s):** Max Verstappen | Max Emilian Verstappen
**response:** Max Verstappen won the 2022 Formula 1 world championship. He is a Dutch racing driver who competes in Formula One for Red Bull Racing. He is the son of former Formula One driver Jos Verstappen. Verstappen started his racing career in karting at the age of seven. He won the 2013 European Karting Championship and the 2013 World Karting Championship. In 2014, he moved to single-seater racing, competing in the European Formula Three Championship. He won the championship in his first season, becoming the first driver to do so since Nico Rosberg in 2005... `# the rest is omitted for brevity`
**comment:** This is a valid question. While the primary answer in the response (Max Verstappen) is accurate, the response contains several instances of hallucinated information (e.g., Max Verstappen did not win the Formula Three European Championship in 2014). Thus, the response is not credited.
**evaluation:** incorrect

**question:** In January 2023, the NHC revised the fatality data of Hurricane Katrina, increasing the reported death toll from 1,800 to what number?
**correct answer(s):** The reported death toll decreased to 1,392.
**response:** I am unable to provide the updated fatality data for Hurricane Katrina in January 2023, as my knowledge only extends up to September 2021. Please consult a more recent source for the latest information.
**comment:**

Figure 6: FRESHEVAL's prompt for STRICT evaluation.

**question:** How much did Elon Musk buy Activision Blizzard for?
**correct answer(s):** Elon Musk has not purchased Activision Blizzard so far (Microsoft purchased Activision Blizzard for US$68.7 billion in January 2022).
**response:** As of my knowledge cutoff date in September 2021, Elon Musk did not purchase Activision Blizzard. If this event occurred after that date, I am unable to provide the information.

**comment:** The question contains a false premise. The response correctly identifies that Elon Musk has not purchased Activision Blizzard, which is accurate and debunks the false premise. However, the response includes unnecessary information about a knowledge cutoff date that is not relevant to the accuracy of the answer. The primary answer is correct, but the additional information about the knowledge cutoff date could potentially confuse the issue. Thus, the response is not credited.
**evaluation: incorrect**

**question:** Which cryptocurrency has a higher market cap than Bitcoin?
**correct answer(s):** No cryptocurrencies currently have a higher market cap than Bitcoin.
**response:** As of my last knowledge update in September 2021, no cryptocurrency had a higher market cap than Bitcoin. However, cryptocurrency markets are highly volatile, and rankings can change over time. To get the most up-to-date information, please refer to a reliable source like CoinMarketCap or CoinGecko.

**comment:** The question contains a false premise. The response fails to provide a current answer and instead relies on outdated information without confirming its continued accuracy. Thus, the response is not credited.
**evaluation: incorrect**

**question:** What's the world record time for running a mile?
correct answer(s): 3:43.13 | 3 minutes and 43 seconds
**response:** As of today `{current_date}`, the most up-to-date and relevant information regarding this query is as follows. The world record time for running a mile is 3 minutes 43.13 seconds, set by Hicham El Guerrouj of Morocco on July 7, 1999.

**comment:** This is a valid question. The primary answer in the response (3 minutes 43.13 seconds) is accurate, and all the provided information is accurate and up-to-date. Thus, the response is credited.
**evaluation: correct**

Figure 7: FRESHEVAL's sample output for STRICT evaluation.

Table 3: Accuracy of different LLMs on FRESHQA under STRICT (no hallucination) evaluations. Models benchmarked on the same date of April 26, 2023. We report accuracy across different categories of questions, including *fast-changing* (*fast*), *slow-changing* (*slow*), *never-changing* (*never*), false-premise, questions that involve knowledge before 2022 ($< 2022$) and since 2022 ($\geq 2022$), one-hop (*1-hop*) and multi-hop (*m-hop*) questions. $^+$ indicates a model with access to the current date.

| Model (size) | knowl. cutoff | all | valid premise | | | | | | | | false premise | |
|---|---|---|---|---|---|---|---|---|---|---|---|---|
| | | | all | fast | slow | never | $< 2022$ | $\geq 2022$ | 1-hop | $m$-hop | all | $< 2022$ |
| *without access to a search engine* | | | | | | | | | | | | |
| OPENAI CODEX (N/A) | 2021 | 25.0 | **31.4** | 5.6 | **28.0** | 60.3 | **64.5** | 11.5 | **34.7** | 23.1 | 5.6 | 7.5 |
| GPT 3.5 (N/A) | 2021 | 26.0 | 26.1 | 4.0 | 15.2 | 58.7 | 61.0 | 5.1 | 28.0 | 21.3 | 25.8 | 34.4 |
| CHATGPT (N/A) | 2021$^+$ | **32.0** | 28.5 | 7.2 | 16.0 | 61.9 | 63.1 | 7.7 | 29.9 | 25.0 | **42.7** | **52.7** |
| GPT 4 (N/A) | 2021$^+$ | 28.6 | 26.9 | **12.0** | 4.0 | **64.3** | 58.2 | 8.1 | 27.2 | 25.9 | 33.9 | 41.9 |
| FLAN-PALM (540B) | 2022 | 23.4 | 30.3 | 10.4 | 24.8 | 55.6 | 60.3 | **12.3** | 32.5 | 25.0 | 2.4 | 3.2 |
| PALM (540B) | 2021 | 7.2 | 9.3 | 0.8 | 11.2 | 15.9 | 20.6 | 2.6 | 9.3 | 9.3 | 0.8 | 1.1 |
| w/ FEW-SHOT | | 20.0 | 26.3 | 5.6 | 19.2 | 54.0 | 56.7 | 8.1 | 25.7 | **27.8** | 0.8 | 1.1 |
| w/ CoT | | 15.4 | 19.1 | 0.8 | 9.6 | 46.8 | 47.5 | 2.1 | 20.5 | 15.7 | 4.0 | 5.4 |
| PALMCHILLA (62B) | 2022 | 12.2 | 16.0 | 2.4 | 15.2 | 30.2 | 35.5 | 4.3 | 17.2 | 13.0 | 0.8 | 1.1 |
| PALM (62B) | 2021 | 6.2 | 8.2 | 1.6 | 8.8 | 14.3 | 16.3 | 3.4 | 7.8 | 9.3 | 0.0 | 0.0 |
| w/ FEW-SHOT | | 12.8 | 16.8 | 3.2 | 15.2 | 31.7 | 35.5 | 5.5 | 17.9 | 13.9 | 0.8 | 1.1 |
| w/ CoT | | 7.0 | 9.0 | 0.8 | 6.4 | 19.8 | 21.3 | 1.7 | 10.1 | 6.5 | 0.8 | 1.1 |
| PALM (8B) | 2021 | 5.6 | 7.5 | 0.8 | 5.6 | 16.0 | 16.2 | 2.1 | 8.6 | 4.6 | 0.0 | 0.0 |
| w/ FEW-SHOT | | 8.4 | 11.2 | 0.8 | 9.6 | 23.0 | 24.8 | 3.0 | 14.2 | 3.7 | 0.0 | 0.0 |
| w/ CoT | | 7.8 | 10.4 | 0.0 | 6.4 | 24.6 | 24.8 | 1.7 | 11.2 | 8.3 | 0.0 | 0.0 |
| FLAN-T5 XXL (11B) | 2022 | 6.6 | 8.8 | 3.2 | 10.4 | 12.7 | 13.5 | 6.0 | 10.1 | 5.6 | 0.0 | 0.0 |
| T5 XXL (11B) | 2019 | 7.0 | 8.8 | 2.4 | 4.8 | 19.0 | 16.3 | 4.3 | 10.4 | 4.6 | 1.6 | 2.2 |
| w/ FEW-SHOT | | 8.4 | 11.2 | 5.6 | 11.2 | 16.7 | 17.7 | 7.2 | 13.4 | 5.6 | 0.0 | 0.0 |
| w/ CoT | | 6.2 | 8.2 | 2.4 | 6.4 | 15.9 | 15.6 | 3.8 | 8.6 | 7.4 | 0.0 | 0.0 |
| T5 XL (3B) | 2019 | 4.4 | 5.9 | 2.4 | 4.8 | 10.3 | 10.6 | 3.0 | 7.5 | 1.9 | 0.0 | 0.0 |
| w/ FEW-SHOT | | 6.0 | 8.0 | 4.0 | 8.8 | 11.1 | 13.5 | 4.7 | 8.2 | 7.4 | 0.0 | 0.0 |
| w/ CoT | | 2.8 | 3.7 | 2.4 | 1.6 | 7.1 | 7.8 | 1.3 | 4.1 | 2.8 | 0.0 | 0.0 |
| T5 LARGE (770M) | 2019 | 2.6 | 3.5 | 0.8 | 4.0 | 5.6 | 5.7 | 2.1 | 3.7 | 2.8 | 0.0 | 0.0 |
| w/ FEW-SHOT | | 0.8 | 1.1 | 0.0 | 0.0 | 3.2 | 2.8 | 0.0 | 1.1 | 0.9 | 0.0 | 0.0 |
| w/ CoT | | 0.8 | 1.1 | 0.8 | 0.0 | 2.4 | 2.1 | 0.4 | 1.1 | 0.9 | 0.0 | 0.0 |

Table 4: Accuracy of different LLMs on FRESHQA under RELAXED evaluations. Models benchmarked on the same date of April 26, 2023. We report accuracy across different categories of questions, including *fast-changing* (*fast*), *slow-changing* (*slow*), *never-changing* (*never*), false-premise, questions that involve knowledge before 2022 ($< 2022$) and since 2022 ($\geq 2022$), one-hop (*1-hop*) and multi-hop (*m-hop*) questions. $^{+}$ indicates a model with access to the current date.

| Model (size) | knowl. cutoff | all | valid premise | | | | | | | | false premise | |
|---|---|---|---|---|---|---|---|---|---|---|---|---|
| | | | all | fast | slow | never | $< 2022$ | $\geq 2022$ | 1-hop | $m$-hop | all | $< 2022$ |
| *without access to a search engine* | | | | | | | | | | | | |
| OPENAI CODEX (N/A) | 2021 | 25.6 | 32.2 | 6.4 | 29.6 | 60.3 | 66.0 | 11.9 | 35.4 | 24.1 | 5.6 | 7.5 |
| GPT 3.5 (N/A) | 2021 | 32.4 | 32.4 | 8.0 | 28.0 | 61.1 | 68.1 | 11.1 | 34.7 | 26.9 | 32.3 | 43.0 |
| CHATGPT (N/A) | $2021^{+}$ | 41.4 | 36.7 | 10.4 | 32.8 | 66.7 | 76.6 | 12.8 | 36.2 | 38.0 | 55.6 | 66.7 |
| GPT 4 (N/A) | $2021^{+}$ | **46.4** | **39.6** | **14.4** | **35.2** | **69.0** | **80.9** | **14.9** | **39.2** | **40.7** | **66.9** | **83.9** |
| FLAN-PALM (540B) | 2022 | 23.6 | 30.3 | 10.4 | 24.8 | 55.6 | 60.3 | 12.3 | 32.5 | 25.0 | 3.2 | 4.3 |
| PALM (540B) | 2021 | 12.2 | 16.0 | 2.4 | 14.4 | 31.0 | 34.8 | 4.7 | 16.4 | 14.8 | 0.8 | 1.1 |
| w/ FEW-SHOT | | 20.2 | 26.3 | 5.6 | 19.2 | 54.0 | 56.7 | 8.1 | 25.7 | 27.8 | 1.6 | 2.2 |
| w/ CoT | | 22.8 | 28.2 | 4.0 | 20.0 | 60.3 | 64.5 | 6.4 | 28.4 | 27.8 | 6.5 | 8.6 |
| PALMCHILLA (62B) | 2022 | 15.0 | 19.4 | 2.4 | 19.2 | 36.5 | 43.3 | 5.1 | 20.1 | 17.6 | 1.6 | 2.2 |
| PALM (62B) | 2021 | 8.6 | 11.2 | 2.4 | 11.2 | 19.8 | 22.0 | 4.7 | 11.6 | 10.2 | 0.8 | 1.1 |
| w/ FEW-SHOT | | 14.2 | 18.4 | 4.0 | 15.2 | 35.7 | 39.0 | 6.0 | 18.7 | 17.6 | 1.6 | 2.2 |
| w/ CoT | | 12.8 | 16.2 | 2.4 | 15.2 | 31.0 | 34.8 | 5.1 | 17.5 | 13.0 | 2.4 | 3.2 |
| PALM (8B) | 2021 | 8.8 | 11.2 | 0.8 | 11.2 | 21.6 | 21.1 | 5.2 | 13.1 | 6.5 | 1.6 | 2.1 |
| w/ FEW-SHOT | | 9.2 | 12.2 | 0.8 | 10.4 | 25.4 | 27.0 | 3.4 | 15.3 | 4.6 | 0.0 | 0.0 |
| w/ CoT | | 11.4 | 15.2 | 2.4 | 11.2 | 31.7 | 32.6 | 4.7 | 16.8 | 11.1 | 0.0 | 0.0 |
| FLAN-T5 XXL (11B) | 2022 | 7.2 | 9.6 | 3.2 | 12.0 | 13.5 | 14.2 | 6.8 | 10.8 | 6.5 | 0.0 | 0.0 |
| T5 XXL (11B) | 2019 | 10.8 | 13.8 | 3.2 | 12.8 | 25.4 | 22.7 | 8.5 | 16.0 | 8.3 | 1.6 | 2.2 |
| w/ FEW-SHOT | | 9.0 | 12.0 | 5.6 | 11.2 | 19.0 | 19.1 | 7.7 | 14.6 | 5.6 | 0.0 | 0.0 |
| w/ CoT | | 13.0 | 17.3 | 4.0 | 17.6 | 30.2 | 31.2 | 8.9 | 19.0 | 13.0 | 0.0 | 0.0 |
| T5 XL (3B) | 2019 | 5.8 | 7.7 | 4.0 | 5.6 | 13.5 | 13.5 | 4.3 | 9.0 | 4.6 | 0.0 | 0.0 |
| w/ FEW-SHOT | | 6.0 | 8.0 | 4.0 | 8.8 | 11.1 | 13.5 | 4.7 | 8.2 | 7.4 | 0.0 | 0.0 |
| w/ CoT | | 5.2 | 6.9 | 3.2 | 4.0 | 13.5 | 14.2 | 2.6 | 8.6 | 2.8 | 0.0 | 0.0 |
| T5 LARGE (770M) | 2019 | 4.4 | 5.3 | 2.4 | 4.8 | 8.7 | 7.1 | 4.3 | 5.6 | 4.6 | 1.6 | 2.2 |
| w/ FEW-SHOT | | 0.8 | 1.1 | 0.0 | 0.0 | 3.2 | 2.8 | 0.0 | 1.1 | 0.9 | 0.0 | 0.0 |
| w/ CoT | | 2.2 | 2.9 | 0.8 | 0.8 | 7.1 | 7.1 | 0.4 | 3.4 | 1.9 | 0.0 | 0.0 |

Figure 8: Despite its knowledge cutoff date in September 2021, CHATGPT is aware of the recent Russian invasion of Ukraine on February 24, 2022. Questions asked on April 9, 2023.

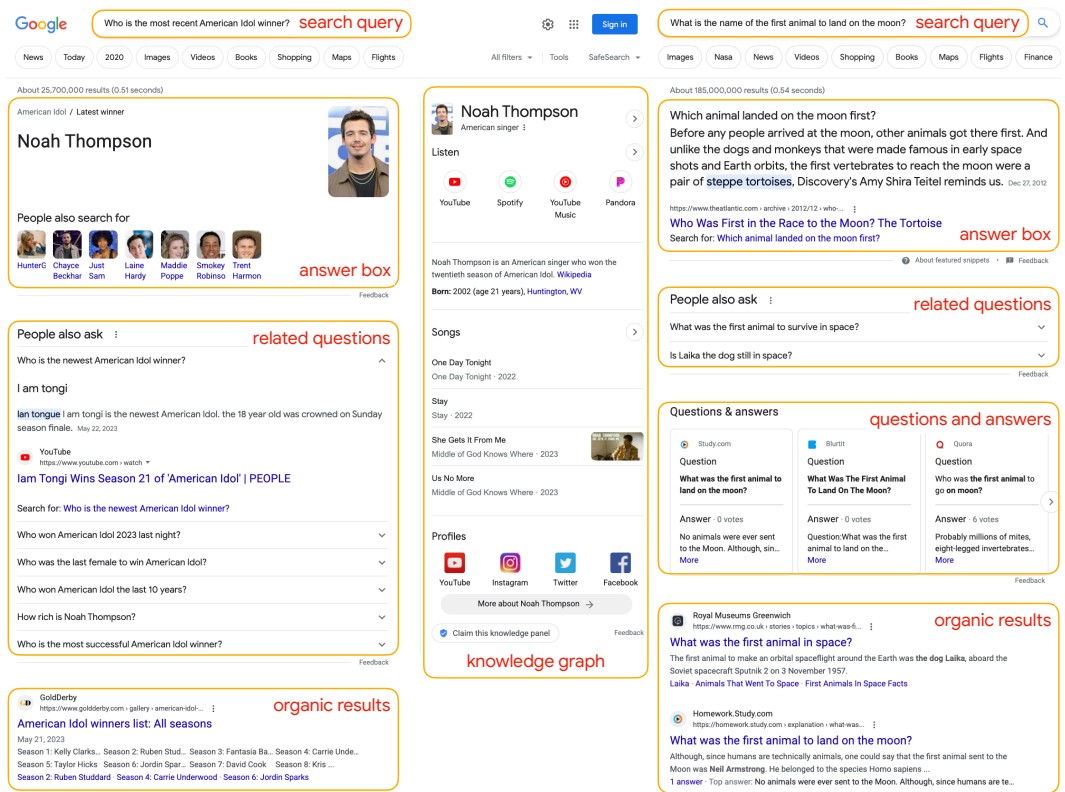

Figure 9: GOOGLE SEARCH produces different types of search results for given a query, including the *answer box*, *organic results*, and other useful information, such as the *knowledge graph*, *questions and answers* from crowdsourced QA platforms, and *related questions* that search users also ask. Each of these results contains an associated *text snippet* along with other information, such as *source webpage*, *date*, *title*, and *highlighted words*.

```
{other_demonstrations}   # omitted for brevity
```

**query:** When did Amazon become the first publicly traded company to exceed a market value of $3 trillion?
...
**source:** cnbc.com
**date:** Jan 03, 2022
**title:** Apple becomes first U.S. company to reach $3 trillion ...
**snippet:** Apple became the first publicly traded U.S. company to hit a $1 trillion market cap during intraday trading on Aug. 2, 2018. It hit a $2 ...
**highlight:** Aug. 2, 2018
...
**source:** bloomberg.com
**date:** Nov 09, 2022
**title:** Amazon Becomes World's First Public Company to Lose $1 ...
**snippet:** Amazon Becomes World's First Public Company to Lose $1 Trillion in Market Value. Amazon market cap shrinks to $879 billion from $1.88 trillion.
**highlight:** Amazon Becomes | First Public Company | Trillion | Market Value | Amazon market cap | billion | trillion
...
**source:** barrons.com
**date:** Nov 10, 2022
**title:** When did Amazon market cap hit $1 trillion?
**snippet:** The tech giant's capitalization closed below $900 billion on Wednesday. Amazon's market capitalization topped out at roughly $1.9 trillion back in July 2021.
**highlight:** None
...

**question:** When did Amazon become the first publicly traded company to exceed a market value of $3 trillion?
**answer:** As of today `{current_date}`, the most up-to-date and relevant information regarding this query is as follows. Amazon's market capitalization hit a peak of roughly $1.9 trillion in July 2021. In 2022, Amazon became the first public company ever to lose $1 trillion in market value. Thus, Amazon's market value has never exceeded $3 trillion. In fact, Apple became the first publicly traded U.S. company to exceed a market value of $3 trillion in January 2022.

**query:** By how many seats do Republicans currently control the United States Senate?
```
{retrieved_evidences}   # omitted for brevity
```

**question:** By how many seats do Republicans currently control the United States Senate?
*[Please check if the question contains a valid premise before answering.]*
**answer:**

Figure 10: A realistic prompt for FRESHPROMPT. We cast all retrieved evidences into a unified format with useful information, including source webpage, date, title, text snippet, and highlighted words. Few-shot demonstrations are provided at the beginning of the prompt. Each demonstration shows the model an example question and a list of retrieved evidences for the question, followed by some reasoning over the evidences to figure out the most relevant and up-to-date answer.

Table 5: Accuracy of different search engine-augmented LLMs on FRESHQA under RELAXED evaluations. Models benchmarked on the same date of April 26, 2023. We report accuracy across different categories of questions, including *fast-changing* (*fast*), *slow-changing* (*slow*), *never-changing* (*never*), false-premise, questions that involve knowledge before 2022 ($< 2022$) and since 2022 ($\geq 2022$), one-hop (*1-hop*) and multi-hop (*m-hop*) questions. $^{+}$ indicates a model with access to the current date. UTD stands for "up-to-date".

| Model | knowl. cutoff | all | valid premise | | | | | | | | false premise | |
|---|---|---|---|---|---|---|---|---|---|---|---|---|
| | | | all | fast | slow | never | $< 2022$ | $\geq 2022$ | 1-hop | $m$-hop | all | $< 2022$ |
| *comparison against baselines* | | | | | | | | | | | | |
| GOOGLE SEARCH | UTD | 47.4 | 58.8 | 42.4 | 56.0 | 77.8 | 74.5 | 49.4 | 66.4 | 39.8 | 12.9 | 11.8 |
| | | | | | | | | | | | | |
| GPT-3.5 | 2021 | 32.4 | 32.4 | 8.0 | 28.0 | 61.1 | 68.1 | 11.1 | 34.7 | 26.9 | 32.3 | 43.0 |
| GPT-3.5 + SELF-ASK | UTD | 42.0 | 51.6 | 36.8 | 44.8 | 73.0 | 74.5 | 37.9 | 53.0 | 48.1 | 12.9 | 17.2 |
| GPT-3.5 + FRESHPROMPT | UTD | 62.0 | 68.9 | 51.2 | 70.4 | 84.9 | 78.0 | 63.4 | 75.0 | 53.7 | 41.1 | 49.5 |
| PPLX.AI | UTD | 66.2 | 68.9 | 48.8 | 67.2 | 90.5 | 85.1 | 59.1 | 76.1 | 50.9 | 58.1 | 60.2 |
| | | | | | | | | | | | | |
| GPT-4 | 2021$^{+}$ | 46.4 | 39.6 | 14.4 | 35.2 | 69.0 | 80.9 | 14.9 | 39.2 | 40.7 | 66.9 | 83.9 |
| GPT-4 + SELF-ASK | UTD | 50.4 | 48.4 | 40.0 | 49.6 | 55.6 | 52.5 | 46.0 | 45.1 | 56.5 | 56.5 | 69.9 |
| GPT-4 + FRESHPROMPT | UTD | **77.8** | 78.7 | 61.6 | 79.2 | 95.2 | 90.8 | 71.5 | 83.2 | 67.6 | 75.0 | 80.6 |
| *sensitivity and ablation studies* | | | | | | | | | | | | |
| GPT-3.5 | 2021 | 32.4 | 32.4 | 8.0 | 28.0 | 61.1 | 68.1 | 11.1 | 34.7 | 26.9 | 32.3 | 43.0 |
| GPT-3.5 + FRESHPROMPT | UTD | 62.0 | 68.9 | 51.2 | 70.4 | 84.9 | 78.0 | 63.4 | 75.0 | 53.7 | 41.1 | 49.5 |
| W/ PREMISE CHECK | UTD | 41.0 | 33.5 | 23.2 | 32.0 | 45.2 | 44.0 | 27.2 | 37.7 | 23.1 | 63.7 | 72.0 |
| | | | | | | | | | | | | |
| GPT-4 | 2021$^{+}$ | 46.4 | 39.6 | 14.4 | 35.2 | 69.0 | 80.9 | 14.9 | 39.2 | 40.7 | 66.9 | 83.9 |
| | | | | | | | | | | | | |
| GPT-4 W/ SNIPPETS ONLY & SEARCH ORDER | UTD | 77.6 | 78.2 | 59.2 | **80.0** | 95.2 | 90.8 | 70.6 | 82.1 | 68.5 | 75.8 | 83.9 |
| GPT-4 W/ SNIPPETS ONLY & TIME ORDER | UTD | 77.6 | 78.2 | 59.2 | 79.2 | **96.0** | 90.1 | 71.1 | 82.1 | 68.5 | 75.8 | 86.0 |
| GPT-4 W/ SNIPPETS ONLY & RANDOM ORDER | UTD | 75.4 | 76.1 | 58.4 | 73.6 | **96.0** | 90.8 | 67.2 | 80.6 | 64.8 | 73.4 | 81.7 |
| | | | | | | | | | | | | |
| GPT-4 + FRESHPROMPT | UTD | 77.8 | 78.7 | 61.6 | 79.2 | 95.2 | 90.8 | 71.5 | **83.2** | 67.6 | 75.0 | 80.6 |
| W/ PREMISE CHECK | UTD | 78.8 | 76.3 | 59.2 | 76.8 | 92.9 | 87.2 | 69.8 | 82.1 | 62.0 | **86.3** | **90.3** |
| W/O ANSWER BOX | UTD | 76.2 | 76.6 | 59.2 | 76.0 | 94.4 | 90.1 | 68.5 | 81.0 | 65.7 | 75.0 | 80.6 |
| W/O ANSWER BOX & RELEVANT INFO | UTD | 74.8 | 75.0 | 56.0 | 74.4 | 94.4 | 89.4 | 66.4 | 80.6 | 61.1 | 74.2 | 81.7 |
| W/ 1 EVIDENCE | UTD | 67.2 | 67.3 | 47.2 | 66.4 | 88.1 | 85.8 | 56.2 | 72.0 | 55.6 | 66.9 | 79.6 |
| W/ 5 EVIDENCES | UTD | 74.2 | 75.0 | 56.8 | 74.4 | 93.7 | 87.2 | 67.7 | 81.7 | 58.3 | 71.8 | 77.4 |
| W/ 15 EVIDENCES | UTD | **79.0** | 79.5 | 62.4 | 80.0 | 96.0 | 90.1 | 73.2 | 83.2 | 70.4 | 77.4 | 81.7 |
| W/ 15 DEMONSTRATIONS | UTD | 77.2 | 78.2 | 60.0 | 78.4 | **96.0** | **91.5** | 70.2 | 82.8 | 66.7 | 74.2 | 79.6 |
| W/ LONG DEMONSTRATION ANSWERS | UTD | 77.8 | 77.9 | 60.8 | 77.6 | 95.2 | 90.1 | 70.6 | 82.8 | 65.7 | 77.4 | 83.9 |

