# OpenReview forum: "FreshLLMs: Refreshing Large Language Models with Search Engine Augmentation"
_ICLR.cc/2024/Conference — Submitted to ICLR 2024_

### Official Review · Reviewer_NG2g · 2023-10-28

**Soundness:** 4 excellent
**Presentation:** 4 excellent
**Contribution:** 3 good
**Rating:** 8
**Confidence:** 4

**Summary:**

In this work the authors conduct detailed study over the factuality and hallucination of text generated by Large Language Models (LLMs) and propose a question answering benchmark FRESHQA which mainly focuses on questions requiring fast-changing world knowledge. Moreover, the authors also present FRESHPROMPT, a few-shot prompting method aiming to boost the performance of LLMs in acquiring up-to-date knowledge retrieved from search engines. Experiments on several LLMs show FRESHPROMPT outperforms other search engine-augmented prompting methods.

**Strengths:**

1.	A novel QA benchmark with 600 questions divided into four main categories: never-changing questions, slow-changing questions, fast-changing questions and false-premise questions is proposed for testing factuality of LLMs. Both questions are generated by nlp researchers and online freelancers and cover a wide range of topics. The authors also commit to update the dataset to get up-to-date knowledge. This benchmark is likely to benefit LLM research a lot.
2.	The paper is well written with clear motivations. A QA benchmark for LLMs is first presented and analysis of different models’ score follows. Then a search engine-based prompting method FRESHPROMPT is proposed to alleviate the problem of factuality, with
3.	Good empirical results. FRESHPROMPT outperforms PERPLEXITY.AI and SELF-ASK on GPT-3.5 and GPT-4. The authors also conduct detailed ablation studies to analyze the results from different perspective.

**Weaknesses:**

1.	The authors conduct experiments on T5, PaLM and GPT series LLMs and show the influence of parameter size on benchmark score. However, I think more experiments on different famous LLMs like LLaMA, Falcon, etc are needed as benchmark baselines.
2.	For better visualization, the best results in Table 1 need to be displayed in bold.

**Questions:**

na

---

> ### Author Response · Authors · 2023-11-22
> **We thank the reviewer for the thoughtful comments**
>
> We thank the reviewer for the thoughtful comments.
>
> 1. Regarding the reviewer’s suggestion about additional experiments, we agree that the paper could be strengthened by providing additional results with recent open-source models. We commit to benchmarking some open models (e.g., Llama 2, Mistral, and Zephyr) in the next version of our paper.
>
>
> 2. We have uploaded a new version of our paper where the best results in tables of results are bolded.

---

### Official Review · Reviewer_4BtD · 2023-10-29

**Soundness:** 3 good
**Presentation:** 3 good
**Contribution:** 3 good
**Rating:** 5
**Confidence:** 3

**Summary:**

The paper analyzes how Large Language Models (LLMs) perform when queried with questions involving fast-changing knowledge or questions requiring the debunking of false premises. To this end, they introduce a new high-quality Q&A dataset, FreshQA, comprising 600 questions. They evaluate the performance of multiple LLMs under two regimes: STRICT, in which an answer is correct if and only if the reasoning and the answer are correct, and RELAXED, in which it is sufficient for the answer to the question to be correct. This sheds light on whether the models can arrive at the correct answer while hallucinating some facts about the world. To improve performance, the authors devise a method to incorporate search results from Google into the context of the LLM. The conclusion is that LLMs struggle with fast-changing and false premise questions (although GPT-4 seems to perform quite well on these types of questions if the knowledge is before its cut-off date), and retrieval-augmented generation improves the performance of LLMs.

**Strengths:**

- Well-written and easy to read
- Provides a high-quality dataset of 600 questions, along with useful question taxonomy (required knowledge dates, is multi-hop, has false premise, etc.)

**Weaknesses:**

- Dataset and paper lacks any type of automatic evaluation metric, everything is based on human eval (major)
- Findings are not particularly novel or unexpected (minor)
- Methodology for retrieval seem to be tailored to Google Search (as the authors acknowledge, minor)

**Questions:**

Thanks to the authors for the thoroughly executed paper. Although I think that the results are not particularly unexpected or novel (e.g. LLMs struggle with multi-hop questions as noted in Self-Ask, or questions that involve fresh, novel and changing knowledge, and that retrieval from a search engine helps), I think the main scientific contribution of this paper might be considered the high-quality dataset of questions.

This leads us to the main weakness of the paper. If I understand correctly, the evaluation in this paper is performed exclusively by human annotation. While I appreciate this substantial evaluation effort, it raises some concerns, especially given that one of the authors' goals is to provide this dataset to the community to facilitate further research in this field.

How many annotators were used to perform each system's evaluation in this paper? Inter-annotator agreement between two evaluators on a subset of 100 questions is reported in the appendix, and it is shown that the agreement is high. Do the authors expect everyone in the research community using the dataset to follow the same evaluation protocol to assess their own systems?

These questions leave me wondering whether the authors could have created an automatic evaluation metric that might make the dataset more broadly useful to the community and less subject to annotator variance.

For example, by carefully crafting the questions with multiple-choice answers (with carefully selected hard negative options that could elicit some of the capabilities the authors are examining, e.g., integrating options with incorrect reasoning but correct answers). Multiple-choice is a strategy already used in well-known QA benchmarks such as MMLU or the suite of the original 11 T0 held-out tasks and lends itself nicely to computing accuracy. Can the authors elaborate on why they didn't choose this option?

Overall, I appreciate the effort put into this paper, but I am a bit unsure about the broad usability of such dataset in the current state. I hope the authors can help me clarify these doubts and, if so, I will be more than willing to increase my score.

---

> ### Author Response · Authors · 2023-11-22
> **We thank the reviewer for the constructive review**
>
> We thank the reviewer for the constructive review. We apologize for the delayed response. We spent the majority of the rebuttal period implementing and evaluating the auto-rater suggested by the reviewer. We are happy to report that it works very reliably and can thus be used in place of human evaluation, and we thank the reviewer for directly improving the accessibility and impact of our FreshQA benchmark.
>
> 1. The reviewer inquired about the number of human raters and whether future work will follow the same evaluation protocol. All model responses underwent a rigorous evaluation process conducted by two authors using our two-mode evaluation protocol. We believe that human evaluators possess the expertise and common sense required to detect issues like hallucinations, making them more reliable than automated evaluation metrics for assessing LLMs' factuality. However, researchers have the flexibility to adjust their evaluation methods if human evaluation proves challenging. An easily implemented alternative is to use standard metrics like F1/exact match or recall, which assess the overlap between the model response and the ground truth answer(s) (e.g., see You.com's recent evaluation: https://about.you.com/introducing-the-you-api-web-scale-search-for-llms/). Alternatively, researchers can utilize LLM-based automatic evaluation metrics such as FactScore (https://arxiv.org/abs/2305.14251) or our newly developed FreshEval metric (details below).
>
>
> 2. The reviewer asked about implementing an auto-rater for FreshQA. To facilitate future evaluations, we have developed FreshEval, a simple automatic metric that uses few-shot in-context learning to teach an LLM to assess model responses, achieving an average agreement of 96.5% with human evaluations for Relaxed and 96% for Strict (please refer to Appendix B in the updated version of our paper).  In each evaluation, the model is conditioned on a given question, a list of valid answers for the question, and a model responses, and is then expected to generate a comment on the correctness of the response, followed by a final judgment (please see Figure 5 and Figure 6 for FreshEval’s prompts, and Figure 7 for its sample output). We are in the process of re-evaluating all models with FreshEval.
>
>
> 3. The reviewer suggested the possibility of reformatting FreshQA into a multiple-choice format, and we appreciate this input. Researchers are encouraged to consider reformatting FreshQA into other formats (e.g., multiple-choice QA, timestamped QA), if it aligns with their specific evaluation needs and objectives. Our dataset serves as an open-ended QA benchmark, allowing us to assess not only the correctness of responses but also the presence of any hallucinations. While adopting a multiple-choice format might streamline evaluation, there are several drawbacks to consider: (1) When presented with multiple-choice questions, models can potentially rely on heuristics to deduce the correct answer rather than genuinely understanding the context; (2) Assessing hallucinations becomes challenging in a multiple-choice setup, as models are simply required to select from predetermined options; and (3) Recent models are designed for open-ended responses, and real-time QA often does not involve presenting multiple choices. Thus, retaining FreshQA's open-ended evaluation can be valuable for assessing models' practical utility in real-world applications.
>
> 4. Regarding the reviewer's comment on the predictability of our results, while it is expected that LLMs without access to real-time data would struggle with questions requiring up-to-date knowledge, our findings on questions with false premises were somewhat unexpected. Regardless of model size, all models had difficulty with such questions, and larger models did not improve accuracy ("flat scaling"). To the best of our knowledge, we are the first to evaluate LLMs at scale on false-premise questions and report this flat-scaling pattern. Additionally, we discovered that LLMs can debunk false-premise questions when explicitly asked.
>
>      On the method side, while it may not be surprising that injecting up-to-date knowledge into LLMs can enhance factual accuracy, our work contributes to the discussion on how to effectively integrate external knowledge, like web search results, into LLMs. We delve into the specifics of optimizing this process, offering practical strategies for maximizing LLM performance. In contrast to previous approaches by Lazaridou et al. (2022) and Press et al. (2022), which use only one retrieved evidence at a time, we demonstrate that recent LLMs can handle an increasing number of retrieved evidences inserted into the prompt simultaneously, significantly boosting performance.
>
> 5. As for FreshPrompt's compatibility with other search engines, we believe it can readily adapt. FreshPrompt's success depends on the number and order of retrieved evidences, which are typically available from most search engines.

---

### Official Review · Reviewer_BRXn · 2023-11-02

**Soundness:** 3 good
**Presentation:** 3 good
**Contribution:** 3 good
**Rating:** 6
**Confidence:** 4

**Summary:**

The paper addresses the challenge of large language models (LLMs) not being updated with current information, leading to inaccuracies in their responses. The authors introduce "FreshQA", a dynamic QA benchmark designed to test the factuality of LLM-generated answers, especially for questions requiring up-to-date knowledge or debunking false premises. Through extensive human evaluations, they highlight the limitations of current LLMs in addressing fast-changing and false-premise questions. To combat this, they propose FreshPrompt, an in-context learning method that augments LLMs with up-to-date information from search engines, significantly enhancing their factuality.

**Strengths:**

1. Tackles a paramount limitation of LLMs – their reliance on outdated or erroneous knowledge.

2. Unveils a dynamic benchmark, FreshQA, capable of evolving over time, which stands as a potent tool for continuous evaluations.

3. Implements a thorough evaluation procedure to gauge both the accuracy and potential hallucination in LLM responses.

**Weaknesses:**

1. The evolving nature of FreshQA could pose challenges for researchers aiming for consistent benchmarks over varied time frames.

2. The FreshQA dataset bears similarities with RealTimeQA and TimeQA, which somewhat dilutes the novelty of this work, although it remains a complementary addition.

3. The proposed method isn't entirely groundbreaking, given precedents like internet-augmented LLM [1] and REPLUG [2].
[1] Internet-augmented language models through few-shot prompting for open-domain question answering
[2] Replug: Retrieval-augmented black-box language models.

**Questions:**

see weakness

---

> ### Author Response · Authors · 2023-11-22
> **We thank the reviewer for the insightful review.**
>
> We thank the reviewer for the insightful review.
>
> 1. The reviewer is concerned about potential challenges posed by FreshQA’s dynamic nature for consistent benchmarks. While FreshQA may be affected by the ever-evolving nature of information, we believe that it remains a valuable resource for the following reasons:
>
>      * First, FreshQA's dynamic nature mirrors real-life scenarios where information is continually updated. This makes it a suitable benchmark for evaluating LLMs’ ability to handle dynamic and ever-changing information or provide answers based on the most recent updates. These evaluations demonstrate their practical utility in real-world applications, where knowledge updates occur in real-time. Researchers are encouraged to use FreshQA for evaluating models that can continuously learn and adapt to new information.
>
>      * Second, researchers can adapt their evaluation methodologies to account for FreshQA's dynamic nature when needed. One approach is to use various “snapshot” versions of FreshQA that freeze answers at specific time points and include timestamps on questions to indicate when they were asked, similar to prior work such as TimeQA and SituatedQA. Alternatively, researchers can assess performance on a subset of the dataset where the relevant information (e.g., question type) has remained consistent over time. These methods help establish stable benchmarks for different time frames, although they may not fully capture the dynamic nature of real-world information.
>
> 2. Regarding the reviewer’s comment on FreshQA’s novelty, below we elaborate on the differences between FreshQA, TimeQA, and RealTime QA, and highlight the unique merits and contributions that FreshQA brings to the field.
>
>      * First, while TimeQA contains mostly synthesized questions, FreshQA exclusively features human-generated natural questions, enhancing the dataset's authenticity and real-world relevance. Additionally, TimeQA is static, with time-stamped questions and fixed answers. Therefore, it does not require a model to access real-time data or browse the Internet for current information. In contrast, FreshQA is a dynamic benchmark where answers may change over time, requiring a model to understand the world's up-to-date knowledge to answer correctly.
>
>      * Second, while RealTime QA emphasizes real-time information updates and how LLMs handle them, it exclusively focuses on news data. It re-evaluates models with approximately 30 new multiple-choice questions about recent events from news websites, annotated weekly by its authors. In contrast, FreshQA features a fixed set of open-ended questions covering a wide range of topics, including arts, music, politics, government, religion, science and technology, environment, transportation, sports, and more. These questions naturally allow for changing answers based on new developments in the world.
>
>      * Third, FreshQA covers a wide variety of question types, including questions that involve old or never-changing world knowledge, questions with false premises that need to be debunked, and single-hop/multihop questions. This diversity of question types is essential for mitigating model bias and preventing overfitting. It guarantees that models' performance remains balanced across different question categories and does not become overly optimized for one category at the expense of others.
>
>      * Lastly, our two-mode evaluation protocol, comprising relaxed and strict evaluations, offers a spectrum for assessing LLMs' factuality. This approach not only provides a deeper and more nuanced understanding of their performance but also quantifies hallucination, a crucial aspect absent in previous QA benchmarks.
>
> 3. The reviewer raised a question about FreshPrompt’s novelty. While FreshPrompt is not the first few-shot prompting method that integrates current information from a search engine into LLMs, it introduces several innovations. These innovations involve fully leveraging a search engine by extracting all up-to-date and relevant information (including knowledge from related questions), and  efficiently handling an increasing number of retrieved evidences (including casting them into a unified format and reasoning over them to identify the most relevant and up-to-date answer). These contributions have resulted in performance improvements. Our analysis provides insights for future work on search engine-augmented LLMs. We found that both the number and order of retrieved evidences significantly impact the correctness of LLM-generated answers. Additionally, instructing the LLM to generate concise and direct answers has been shown to reduce hallucination.
>
>      Compared to Lazaridou et al. (2022)'s approach, which involves using a single retrieved evidence at a time and requires 50 inference calls to an LLM for each question to generate multiple candidate answers and select the best one, FreshPrompt is more efficient, requiring only a single inference call to an LLM.

---

### Meta-Review · Area_Chair_ZUYu · 2023-12-06

**Metareview:**

This paper introduces a new dynamic question answering benchmark, FreshQA, that aims to assess models' ability to answer questions on current world knowledge. The work assesses a set of LLMs on the benchmark, demonstrating the limitations of these models for such dynamic knowledge. The paper also includes experiments with a new few-shot prompting method, FreshPrompt that incorporates recent relevant information in the prompt, and demonstrates that this helps improve performance in comparison to baselines, such as Self-Ask. The work is interesting and useful, however, a major limitation is the lack of an automated metric, as raised by the reviewers (and this wasn't sufficiently considered in the rebuttal).

**Justification For Why Not Higher Score:**

The work is interesting, however the contributions are limited.

**Justification For Why Not Lower Score:**

N/A

---

### Decision · Program_Chairs · 2024-01-16

Reject